# Expanding the Molecular Spectrum of *MMP21* Missense Variants: Clinical Insights and Literature Review

**DOI:** 10.3390/genes16010062

**Published:** 2025-01-08

**Authors:** Domizia Pasquetti, Paola Tesolin, Federica Perino, Stefania Zampieri, Marco Bobbo, Thomas Caiffa, Beatrice Spedicati, Giorgia Girotto

**Affiliations:** 1Institute for Maternal and Child Health—IRCCS “Burlo Garofolo”, 34137 Trieste, Italy; domizia.pasquetti@burlo.trieste.it (D.P.); paola.tesolin@burlo.trieste.it (P.T.); stefania.zampieri@burlo.trieste.it (S.Z.); marco.bobbo@burlo.trieste.it (M.B.); thomas.caiffa@burlo.trieste.it (T.C.); giorgia.girotto@burlo.trieste.it (G.G.); 2Department of Medicine, Surgery and Health Sciences, University of Trieste, 34149 Trieste, Italy; federica.perino@burlo.trieste.it

**Keywords:** *MMP21*, Heterotaxy, Congenital Heart Defects

## Abstract

Background/Objectives: The failure of physiological left-right (LR) patterning, a critical embryological process responsible for establishing the asymmetric positioning of internal organs, leads to a spectrum of congenital abnormalities characterized by laterality defects, collectively known as “heterotaxy”. *MMP21* biallelic variants have recently been associated with heterotaxy syndrome and congenital heart defects (CHD). However, the genotype–phenotype correlations and the underlying pathogenic mechanisms remain poorly understood. Methods: Patients harboring biallelic *MMP21* missense variants who underwent diagnostic genetic testing for CHD or heterotaxy were recruited at the Institute for Maternal and Child Health—I.R.C.C.S. “Burlo Garofolo”. Additionally, a literature review on *MMP21* missense variants was conducted, and clinical data from reported patients, along with molecular data from in silico and modeling tools, were collected. Results: A total of 18 *MMP21* missense variants were reported in 26 patients, with the majority exhibiting CHD (94%) and variable extra-cardiac manifestations (64%). In our cohort, through Whole-Exome Sequencing (WES) analysis, the missense p.(Met301Ile) variant was identified in two unrelated patients, who both presented with heterotaxy syndrome. Conclusions: Our comprehensive analysis of *MMP21* missense variants supports the pathogenic role of the p.(Met301Ile) variant and provides significant insights into the disease pathogenesis. Specifically, missense variants are distributed throughout the gene without clustering in specific regions, and phenotype comparisons between patients carrying missense variants in compound heterozygosity or homozygosity do not reveal significant differences. These findings may suggest a potential loss-of-function mechanism for *MMP21* missense variants, especially those located in the catalytic domain, and highlight their critical role in the pathogenesis of heterotaxy syndrome.

## 1. Introduction

Although humans exhibit external mirror symmetry between the left and right sides of the body, the internal organs are asymmetric in their placement. Left-right (LR) patterning is an embryological process that relies on the precise timing of transcription factor activation, occurring after the establishment of the anterior-posterior and dorsal-ventral axes [1]. Although the biological mechanisms underlying LR patterning remain largely unknown, the roles of both motile and non-motile cilia in the process are well established. These cellular structures are essential to the function of a transient embryonic structure known as the Left-Right Organizer (LRO), which becomes visible in human embryos around 14 days post-fertilization [2]. The motile cilia are tilted toward the posterior and rotate clockwise, producing a leftward fluid flow [3]. This induces asymmetric calcium transients in the non-motile cilia of the LRO and ultimately results in the left-sided expression of key transcription factors in the lateral plate mesoderm, leading to the establishment of normal organ asymmetry [4].

A failure to properly develop LR patterning results in a spectrum of congenital abnormalities characterized by defects in laterality, collectively referred to as “heterotaxy” [5]. Normal lateralization is termed “situs solitus”, while heterotaxy comprehends “situs inversus totalis”, a mirror-image lateralization defect, and “situs ambiguus”, a spectrum of laterality defects occurring between situs solitus and situs inversus totalis. Heterotaxy, with a prevalence of 0.001% in live births [6], is characterized by great phenotypic heterogeneity and may include congenital heart defects (CHD), intestinal rotational abnormalities, impaired splenic function, respiratory symptoms and other anomalies. CHD occurs in approximately 80% of individuals with heterotaxy, reflecting the well-proven key role of LR signaling in the embryologic development of the heart. The heterotaxy-associated CHD spectrum may include anomalies of the pulmonary and systemic venous return, atrioventricular septal defects, ventricular inversion, transposition of the great arteries, double-outlet right ventricle, and heart rhythm disorder [7].

The genetic architecture of heterotaxy is complex and its investigation has been complicated by its rarity and phenotypic heterogeneity; however, chromosomal copy number variants (CNV) [8] and single-nucleotide variants (SNV) in disease-causing genes have been identified in a low percentage of isolated forms of heterotaxy [9]. Also, syndromic forms of heterotaxy have been recognized, more frequently in the spectrum of Primary Ciliary Dyskinesia—caused by pathogenetic variants (PVs) in the genes responsible for motile cilia structure and function [10,11]—or Ciliopathies, disorders affecting both motile and immotile cilia [12].

## 2. Materials and Methods

Patients with a prenatal or postnatal diagnosis of heterotaxy syndrome who underwent diagnostic genetic testing were recruited at the Institute for Maternal and Child Health—I.R.C.C.S. “Burlo Garofolo” (Trieste, Italy). For this study, patients harboring biallelic *MMP21* missense variants were subsequently selected.

Genetic testing included Whole exome sequencing (WES), which was performed on an Illumina NextSeq550 system (Illumina Inc., San Diego, CA, USA) using the Twist Human Exome 2.0 Plus Comprehensive Exome Spike-in kit (Twist Bioscience, San Francisco, CA, USA), according to the manufacturer’s protocol. WES data were analyzed using the EnGenome Expert Variant Interpreter (eVai) software v.3.4 (https://evai.engenome.com, accessed on 10 December 2024) for reads alignment, variant calling, and annotation. American College of Medical Genetics (ACMG) guidelines [13] were then applied for genomic variant classification and prioritization.

A literature review was performed using PubMed, and papers containing the keywords “MMP21” and “heterotaxy” were extracted. All *MMP21* missense variants reported in the literature were collected, regardless of their ACMG classification.

MMP21 protein domains were defined employing SMART [14], while MMP21 protein tridimensional structure was retrieved from the AlphaFold Protein Structure Database [15]. Several in silico prediction tools were employed to predict the pathogenicity of the missense variants, including AlphaMissense [16], SIFT [17], Polyphen [18], CADD [19], and DANN [20]. Finally, variants frequencies were defined according to the GnomAD database [21] and and current ACMG classification was verified using the available literature and the ClinVar database [22]. Amino acid conservation across species of Met301 residue was investigated by protein multiple-sequence alignments with Clustal Omega (https://www.ebi.ac.uk/jdispatcher/msa/clustalo, accessed on 10 December 2024) [23].

## 3. Results

### 3.1. Patients Carrying Missense MMP21 Variant

Patient 1, a 14-month-old boy and first child of healthy consanguineous parents (Figure 1), was born following a pregnancy complicated by fetal malformations identified during the second-trimester ultrasound, described as situs inversus with right-sided stomach, left-sided gallbladder, and a complex congenital heart defect. His family history included a reported heart murmur in the father, which spontaneously resolved in early infancy and never required specialist evaluation, as well as late-onset valvulopathy in an uncle on the paternal side. He was delivered at 36 + 3 weeks of gestational age, by caesarean section due to fetal bradycardia. At birth, heterotaxy syndrome was confirmed through a cardiological evaluation, which revealed dextrocardia, common atrium with complete absence of interatrial septum, inferior vena cava interruption, persistent left superior vena cava, a complete atrioventricular septal defect, double-outlet right ventricle with subvalvular and valvular pulmonary stenosis and right-sided aortic arch. The infant also showed an atrial rhythm with bradycardia at 90 bpm during daytime and 65 bpm when sleeping. After palliation with stenting of ductus arteriosus and pulmonary valvuloplasty at 1 month, the infant underwent bilateral cavopulmonary anastomosis (Glenn operation) with separation of the main pulmonary artery from the heart (blood flow from the systemic returns proceeds directly to the lungs, then to the common atrium and ventricles, and then to the unique outflow, the aorta). The patient also received an epicardial pacemaker for severe bradycardia. At first genetic evaluation (14 months of age), the patient exhibited normal global neurodevelopment and auxological parameters within the normal range. No major dysmorphic features or other signs of a syndromic form of congenital heart disease were observed. WES analysis revealed the homozygous c.903G>A, p.(Met301Ile) variant in the *MMP21* gene (NM_147191.1) in the proband. Subsequently, familial segregation confirmed the inheritance from healthy carrier parents.

Patient 2, a 12-year-old girl and second-born of healthy non-consanguineous parents (Figure 1), was born at term by eutocic delivery after an uneventful pregnancy. Auxological parameters at birth were reported as normal. Family history was unremarkable for neurodevelopmental disorders, epilepsy or congenital malformations. At 4 months, she experienced two episodes of seizures, characterized by loss of consciousness, eversion of the head and eyes, tonic-clonic movements of the left hemisome and subsequent generalized seizure. An electroencephalogram showed a pathological pattern. Seizures were refractory to valproate treatment, occurring at a frequency of 1–2 times per day until 18 months of age, when she experienced a status epilepticus. Anti-seizure treatment was escalated with the addition of both valproate and clonazepam, but seizures persisted at a frequency of 3–5 times per week. At 7 years of age, she was hospitalized for drug-resistant epilepsy. Treatment was successfully adjusted to include daily administration of primidone, levetiracetam, clonazepam, and carbamazepine. Brain MRI and CT were performed and returned normal results. Motor milestones were reached at a normal age; however, motor impairment, and frequent falls and stumbles were noted from the age of 3 years. She also experienced a slight language delay: She spoke her first words at 14 months with subsequent language stagnation and slow normal language acquisitions. At age 6, during a cardiological evaluation, she was diagnosed with congenitally-corrected transposition of the great arteries. In this condition, connections between ventricles and atria are discordant, so the right ventricle is in systemic position with consequent pressure overload that leads to ventricular dilatation and dysfunction. Moreover, the patient showed severe pulmonary stenosis. Genetic evaluation at the age of 8 years showed normal growth parameters, scoliosis, diffuse hypertrichosis, and two supernumerary nipples; no major dysmorphisms were noted. First-line genetic testing included SNP-array analysis, negative for aneuploidies or pathogenetic copy number variations (CNV); subsequently, WES identified a heterozygous c.2591_2593delTGC, p.(Leu864del) variant in the *SCN1A* (NM_001165963.1) gene, present de novo in the proband and causative of the epileptic phenotype, and a homozygous c.903G>A, p.(Met301Ile) variant in the *MMP21* (NM_147191.1) gene, inherited from healthy parents. WES results were therefore compatible with a dual molecular diagnosis, which, according to recent literature, is not a rare event, accounting for approximately 5–10% of molecularly solved cases [24].

### 3.2. Clinical and Molecular Spectrum of MMP21 Missense Variants

Literature review of *MMP21* variants showed a total number of 17 additional previously-reported missense variants (Table 1). They are located in all exons, except exon 3, and in all protein domains (Figure 2A,B). All variants are reported with low frequency in healthy population and never found in homozygosity, with the exception of c.163C>T, p.(Arg55Trp) and c.101C>T, p.(Ser34Leu), with allele frequencies of 3.1550% and 0.2752%, respectively. Moreover, variants are predicted to be damaging by the majority of in silico prediction tools (Table 1).

*MMP21* missense variants were reported in 26 patients, including our two cases, with the majority exhibiting CHD (94%). Extra-cardiac manifestations were documented in 15 out of 26 patients (64%). Heterotaxy was diagnosed via prenatal ultrasound in 6 patients, 5 of whom underwent pregnancy termination. Positive outcomes, defined as surgical correction achieving good hemodynamic heart function, were observed in half of patients diagnosed postnatally (Table 2).

## 4. Discussion

*MMP21* encodes for a member of the matrix metalloproteinase (MMPs) family, which collectively act as zinc-dependent endopeptidases. MMPs are critical for both physiological processes and pathological conditions. Indeed, in vitro modeling showed that extracellular matrix degradation, mediated by MMPs, is required for human vasculogenesis [34] and morphogenesis [35]. Among this group, MMP21 protein contains a catalytic metalloproteinase domain and hemopexin domains in the C-terminal. *MMP21* spans 9.3 Kb, located in 10q26.2, and it consists of seven coding exons [36].

The p.(Met301Ile) missense variant involves a well-conserved amino acid residue in the catalytic domain (Figure 2D). It is an unreported variant in ClinVar; also, it is predicted to be damaging by AlphaMissense, SIFT, PolyPhen, CADD, and DANN. Indeed, methionine and isoleucine show distinct properties: The first is a weakly polar amino acid that contains a sulfur atom, while isoleucine is highly hydrophobic, with a rigid and branched structure which is needed to stabilize the hydrophobic core of proteins [37].

The role of *MMP21* in human disease was first described by Guimier et al. in a cohort of eleven families, including nine index cases of heterotaxy, all carrying biallelic *MMP21* variants, supported by functional validation in Mmp21-mutant mice and mmp21-morphant zebrafish [25]. Subsequently, additional patients were identified in larger cohorts of individuals with a post-natal diagnosis of CHD or heterotaxy [26,27,28,30,31]. Interestingly, the use of WES in prenatal diagnosis led to the identification of *MMP21* variants in six fetuses [29,32,33], suggesting that *MMP21* biallelic variants are a frequent cause of fetal congenital malformation and that the gene should be included in next-generation sequencing (NGS) panels for CHD.

However, the main limitations of the available literature are the lack of reliable *MMP21* genotype–phenotype correlations and the insufficient patient phenotyping in some single-patient case reports. Moreover, the clinical association of missense variants with specific sub-phenotypes has not been systematically studied to date. Our review demonstrates that patients carrying biallelic *MMP21* variants of which at least one is missense invariably exhibit CHD while extra-cardiac involvement varies among patients, including intestinal malrotation, polysplenia, left-sided liver, and right-sided stomach. From a molecular perspective, the absence of variant clustering in specific domains suggests that alterations of the MMP21 protein structure are poorly tolerated. Notably, missense variants are observed with similar frequencies in compound heterozygosity (46%) and in homozygosity (52%). In the first group, two patients were compound heterozygous for distinct missense variants [p.(Arg31Trp); p.(Glu215Lys) and p.(Ser186Ile); p.(Ala184Val), respectively], while the remaining patients carried one loss-of-function variant, primarily stop-gain or frameshift mutations. Among these cohorts, it is worth noting the presence of five sibling pairs, three carrying a homozygous *MMP21* missense variant and two carrying compound heterozygous *MMP21* missense variants. However, we did not detect a significant difference in the associated phenotypes, either among unrelated individuals or within the sibling pairs. These clinical observations lead to the speculation that missense variants likely do not exert a hypomorphic effect on the protein. Functional validation studies of *MMP21* missense variants were previously conducted for three heterozygous variants [p.(Gly244Glu), p.(Leu277Phe), and p.(Lys487Glu)] identified in three unrelated patients with complex CHDs [38]. Based on the normal mRNA and protein expression in mutant HEK-293T cells, as well as zebrafish mRNA rescue assays and in silico analyses, the authors proposed that the p.(Gly244Glu) and p.(Lys487Glu) variants may function as dominant negatives, thereby acting as predisposing genetic risk factors in heterozygosity. Although the exact role of monoallelic *MMP21* variants in human disease remains to be conclusively demonstrated, it is noteworthy that none of the previously reported variants were identified in our cohort.

Our study provides evidence supporting the pathogenetic role of the *MMP21* p.(Met301Ile) missense variant, which was identified in two unrelated patients who both presented with complex heterotaxy syndrome. For the first time, we have comprehensively reviewed and analyzed the clinical and molecular features of *MMP21* missense variants. Given the phenotypes observed in patients carrying *MMP21* missense variants, we hypothesize that their pathogenic mechanism may involve a direct impact on the structural or functional integrity of the protein, mimicking a loss-of-function effect, particularly when the variants are located within the critical zinc-binding domain. Nevertheless, functional studies are still required to confirm the molecular mechanisms underpinning these observations. Since *MMP21* gene expression is high in fetal tissues and low in different adult tissues [39], the use of patient-derived specimens or biopsies is complex and therefore cell-based or animal model studies may be more appropriate. Regarding the potential functional validation of the loss-of-function effects of *MMP21* missense variants, one approach could involve enzymatic assays to evaluate the hypothesized absence or significant reduction of proteolytic activity on the extracellular matrix. These results should subsequently be compared to the reduction observed for *null* alleles. Such studies would provide additional insights into the precise biological consequences of *MMP21* missense variants, improve the classification of *MMP21* missense variants of unknown significance and shed light on possible genotype–phenotype correlations. All these findings would be extremely relevant for providing more accurate molecular diagnoses and therapeutic strategies for heterotaxy patients.

## Figures and Tables

**Figure 1 genes-16-00062-f001:**
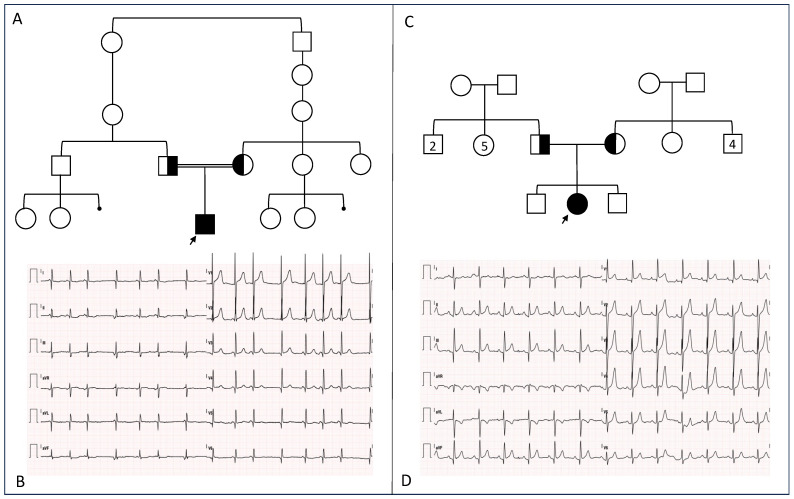
(**A**) Family pedigree of Patient 1, showing grade of parents’ consanguinity, (**B**) ECG of Patient 1 displaying multifocal atrial beats with junctional rhythm, right axis deviation and positive T wave in right precordial leads; (**C**) Family pedigree of Patient 2, (**D**) ECG of Patient 2 showing suggestive findings of ventricular inversion, including absence of q waves in V5–V6 and presence of q waves in V1. Filled symbols with black arrows represent index patients; half-filled symbols represent healthy carrier individuals; numbers in pedigree represent the number of individuals with the same degree of biological relationship (e.g., Patient 2 has two paternal uncles).

**Figure 2 genes-16-00062-f002:**
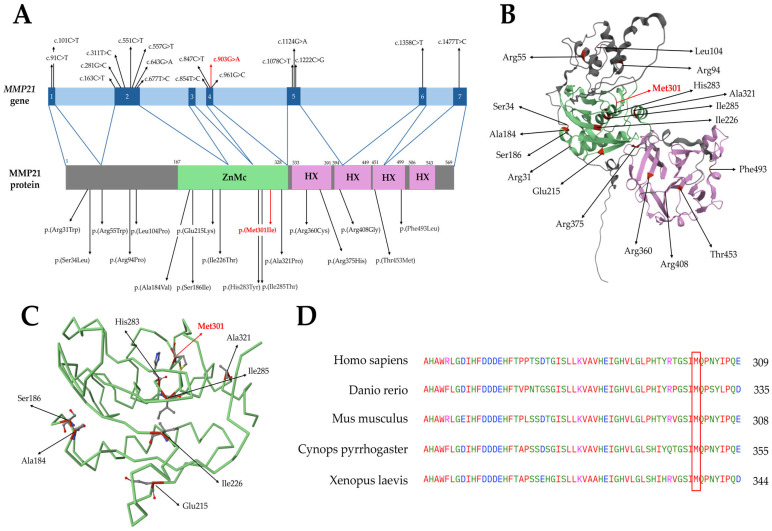
(**A**) Schematic representation of the *MMP21* gene and the MMP21 protein. Exons are depicted in blue and introns in light blue. Protein domains are represented in different colors and aminoacidic positions at the beginning and end of each domain are reported. Missense variants identified in the literature are reported. (**B**) Structural model of MMP21 as predicted by AlphaFold (XZ plane). Protein backbone is represented in gray, while domains are reported in the same color as Figure 2A (i.e., ZnMc domain in green and HX domains in pink). Aminoacidic positions in which missense variants occur are colored in red in the 3D structure. (**C**) Backbone of the ZnMc domain of MMP21 is reported in green. Positions in which missense variants occur are colored in red and amino acid structures are highlighted (carbon atoms are depicted in grey, oxygen in red, nitrogen in blue, and sulfur in yellow). (**D**) Protein alignment showing conservation of methionine 301 across species, highlighted by the red frame; amino acids with similar physicochemical properties are represented in the same colors, as defined by the Clustal Omega software. (ZnMc: Zinc-dependent metalloprotease; HX: Hemopexin-like repeats). In figures, the variant identified in our patients, c.903G>A, p.(Met301Ile), is reported in red.

**Table 1 genes-16-00062-t001:** Eighteen missense variants identified within the *MMP21* gene (NM_147191.1). Chromosome position refers to the build GRCh38. Variant identified in our patients is highlighted in red. (CH: compound heterozygosity; HOM: Homozygous; N/A: Not Available; B: benign; LP: likely pathogenic; US: unknown significance; P: Pathogenic; D: damaging; T: tolerated; PD: probably damaging).

Variant	Amino Acid Substitution	Chromosome Position	Zygosity	Allele Frequency	Clinvar	AlphaMissense	SIFT	PolyPhen	CADD	DANN
c.91C>T	p.(Arg31Trp)	chr10:125,775,731	1 CH	0.0003%	N/A	0.188 (B)	D (0)	PD (0.997)	D (27.5)	D (0.999)
c.101C>T	p.(Ser34Leu)	chr10:125,775,721	1 CH	0.2752%	B	0.157 (B)	D (0.02)	B (0.196)	D (22.3)	D (0.999)
c.163C>T	p.(Arg55Trp)	chr10:125,774,365	1 CH	3.1550%	N/A	0.142 (B)	D (0.01)	PD (0.822)	D (21.5)	D (0.999)
c.281G>C	p.(Arg94Pro)	chr10:125,774,242	1 CH	0.0011%	LP	0.657 (LP)	D (0.01)	PD (0.577)	D (20.9)	D (0.987)
c.311T>C	p.(Leu104Pro)	chr10:125,774,217	2 HOM	0.0002%	N/A	0.256 (B)	D (0.01)	D (1)	D (24.6)	D (0.996)
c.551C>T	p.(Ala184Val)	chr10:125,773,977	1 CH	0.0002%	N/A	0.260 (B)	D (0)	B (0.369)	D (20.6)	D (0.999)
c.557G>T	p.(Ser186Ile)	chr10:125,773,971	1 HOM + 1 CH	0.0000%	LP	0.924 (LP)	D (0)	PD (0.99)	D (24.5)	D (0.997)
c.643G>A	p.(Glu215Lys)	chr10:125,773,885	2 HOM + 1 CH	0.0174%	US	0.610 (LP)	D (0)	PD (0.985)	D (25.3)	D (0.999)
c.677T>C	p.(Ile226Thr)	chr10:125,773,851	2 CH	0.0036%	P	0.975 (LP)	D (0)	PD (0.947)	D (25)	D (0.997)
c.847C>T	p.(His283Tyr)	chr10:125,772,350	2 CH	0.0001%	P	0.982 (LP)	D (0)	D (1)	D (27.8)	D (0.998)
c.854T>C	p.(Ile285Thr)	chr10:125,772,343	1 CH	0.0044%	P	0.738 (LP)	D (0)	D (1)	D (25.3)	D (0.997)
c.903G>A	p.(Met301Ile)	chr10:125,772,294	2 HOM	0.0018%	N/A	0.990 (LP)	D (0)	D (1)	D (25.1)	D (0.997)
c.961G>C	p.(Ala321Pro)	chr10:125,772,236	1 HOM	0.0001%	P	0.898 (LP)	D (0)	PD (0.993)	D (25.1)	D (0.998)
c.1078C>T	p.(Arg360Cys)	chr10:125,770,493	2 HOM	0.0007%	US	0.274 (B)	D (0.01)	PD (0.981)	D (31)	D (0.999)
c.1124G>A	p.(Arg375His)	chr10:125,770,447	3 HOM	0.0005%	N/A	0.270 (B)	T (0.19)	D (1)	D (26.9)	D (0.999)
c.1222C>G	p.(Arg408Gly)	chr10:125,770,349	1 CH	0.0004%	N/A	0.283 (B)	T (0.25)	PD (0.872)	D (22)	D (0.988)
c.1358C>T	p.(Thr453Met)	chr10:125,767,584	1 HOM	0.0095%	US	0.162 (B)	D (0)	PD (0.98)	D (24.1)	D (0.999)
c.1477T>C	p.(Phe493Leu)	chr10:125,766,895	1 HOM	0.0012%	N/A	0.929 (LP)	D (0)	PD (0.816)	D (23.5)	D (0.998)

**Table 2 genes-16-00062-t002:** Cardiac and extra-cardiac phenotype of 26 reported patients carrying *MMP21* missense variants (TOP: termination of pregnancy; Dx: dextrocardia; IVC: inferior vena cava interruption; PLSVC: persistent left superior vena cava; AVSD: atrioventricular septal defect; ASD: atrial septal defect; VSD: ventricular septal defect; DORV: double-outlet right ventricle; TGA: transposition of the great arteries; cc-TGA: congenitally corrected transposition of the great arteries; y.o.: years old; mo.: months; d.: days; N/A: not available). In the column “*MMP21* variants,” patients’ genotypes are described as follows: If a single variant is present, genotype is homozygous; if two variants are present, they are in a compound heterozygous state.

Patients [Reference]	Age at Diagnosis	Outcome	Cardiac Anomalies	Extra-Cardiac Laterality Defects	*MMP21* Variants
P1 [present study]	Prenatal	Positive	Dx, common atrium with complete ASD, IVC, PLSVC, complete AVSD, DORV with subvalvular and valvular pulmonary stenosis, right-sided aortic arch	None	c.903G>A; p.(Met301Ile)
P2 [present study]	Pediatric (6 y.o.)	Positive	cc-TGA and severe pulmonary stenosis	None	c.903G>A; p.(Met301Ile)
P3 [25]	Prenatal	TOP	IVC with azygous continuation, partial anomalous pulmonary venous return, AVSD, cleft anterior mitral valve leaflet, hypoplastic left ventricle, Dx	Intestinal malrotation, polysplenia	c.677T>C; p.(Ile226Thr)
					c.1203G>A; p.(Trp401*)
P4 [25]	Postnatal	N/A	Left superior vena cava draining to coronary sinus, AVSD, abnormal atrioventricular connection, right aortic arch with mirror image branching, patent ductus arteriosus	Abdominal situs ambiguus	c.677T>C; p.(Ile226Thr)
					c.1203G>A; p.(Trp401*)
P5 [25]	Postnatal	Positive	Bilateral superior vena with bridging vein, IVC with azygos continuation, hepatic veins to bilateral atriums, mitral atresia, single ventricle.	Left pulmonary isomerism, left-sided liver, right-sided stomach, polysplenia.	c.91C>T; p.(Arg31Trp)
					c.643G>A; p.(Glu215Lys)
P6 [25]	Postnatal	N/A	Common atrium, complete atrioventricular canal defect, TGA	Thoracic situs ambiguus, midline liver, intestinal malrotation	c.961G>C; p.(Ala321Pro)
P7 [25]	Postnatal	Positive	TGA with VSD, valvar pulmonary stenosis	Thoracic and abdominal situs ambiguus	c.1078C>T; p.(Arg360Cys)
P8 [25]	Postnatal	Positive	Partial anomalous pulmonary venous return, secundum ASD, perimembranous VSD	Thoracic and abdominal situs ambiguus	c.1078C>T; p.(Arg360Cys)
P9 [25]	Postnatal	N/A	Bilateral superior vena cava with no bridging vein, IVC with azygous continuation, partial anomalous pulmonary venous return, patent foramen ovale, inlet and perimembranous VSD, tricuspid regurgitation	Abdominal situs ambiguus	c.1124G>A; p.(Arg375His)
P10 [25]	Postnatal	N/A	None	Situs inversus totalis	c.1124G>A; p.(Arg375His)
P11 [25]	Postnatal	N/A	N/A	N/A	c.1124G>A; p.(Arg375His)
P12 [25]	N/A	N/A	Common atrium, Dx	None	c.1222C>G; p.(Arg408Gly)
					c.1585_1588dup; p.(Val530Glyfs*3)
P13 [25]	N/A	N/A	Common atrium, TGA, pulmonary artery atresia	Thoracic situs inversus	c.101C>T; p.(Ser34Leu)
					c.1372C>T; p.(Arg458*)
P14 [25]	Postnatal	Positive	TGA with VSD, hypoplastic right ventricle, pulmonary artery atresia, peripheral pulmonary artery stenosis, Dx	None	c.163C>T; p.(Arg55Trp)
					c.1372C>T; p.(Arg458*)
P15 [26]	N/A	N/A	Dx, right atrial isomerism, TGA, DORV	None	c.311T>C; p.(Leu104Pro)
P16 [26]	N/A	N/A	Dx, right atrial isomerism, TGA, DORV	None	c.311T>C; p.(Leu104Pro)
P17 [27]	N/A	N/A	Dx, TGA, Abnormal Aortic Arch	Abdominal situs inversus	c.643G>A; p.(Glu215Lys)
P18 [27]	N/A	N/A	Arterial Malposition, Single Ventricle	Abdominal situs inversus	c.557G>T; p.(Ser186Ile)
P19 [28]	Neonatal (6 mo.)	Positive	VSD, Left malposed great arteries, severe tricuspid valve regurgitation, DORV	None	c.1477T>C, p.(Phe493Leu)
P20 [29]	Prenatal	TOP	Pulmonary atresia, univentricular heart, VSD	Right-sided stomach	c.1372C>T, p.(Arg458*)
					c.281G>C, p.(Arg94Pro)
P21 [30]	Prenatal	TOP	Total anomalous venous return, mitral atresia, DORV	N/A	c.947G>A; p.(Trp316*)
					c.847C>T; p.(His283Tyr)
P22 [30]	Neonatal (8 mo)	Positive	Dx, left atrial isomerism, IVC, Large sinus venosus type ASD, Large subaortic VSD, Type B interrupted aortic arch	Tracheomalacia and right main bronchomalacia	c.947G>A; p.(Trp316*)
					c.847C>T; p.(His283Tyr)
P23 [30]	Pediatric (6 y.o.)	Positive	Left atrial isomerism, biventricular atrioventricular connection, morphological left ventricle to right side, discordant ventriculoarterial connection, IVC with azygous continuation, small perimembranous VSD.	None	c.1380_1381delGA; p.(Lys461Valfs*14)
					c.854T>C; p.(Ile285Thr)
P24 [31]	Neonatal (10 d.)	Positive	Congenital heart defects	Midline liver	c.1358C>T, p.(Thr453Met)
P25 [32]	Prenatal	TOP	Common atrium, mitral atresia, hypoplastic left ventricle, VSD, right pulmonary isomerism	Right pulmonary isomerism, intestinal malrotation	c.186del, p.(Trp62CysfsTer48)
					c.643G>A, p.(Glu215Lys)
P26 [33]	Prenatal	TOP	Dx, Pulmonary artery atresia, VSD	N/A	c.557G>T, p.(Ser186Ile)
					c.551C>T, p.(Ala184Val)

## Data Availability

The original data presented in the study are openly available in the main text.

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
