# Peer review of "Expanding the Molecular Spectrum of MMP21 Missense Variants: Clinical Insights and Literature Review"

_genes, 2025, doi:10.3390/genes16010062_

Round 1
Reviewer 1 Report
Comments and Suggestions for Authors
The paper is well-executed and presents clinically significant findings with strong methodological support. However, it lacks functional validation and deeper critical analysis of reviewed data. Addressing these limitations will enhance the paper's robustness and impact. The paper addresses the clinical and molecular aspects of MMP21 missense variants in heterotaxy and congenital heart defects (CHD), a rare but significant condition. The study fills a gap by correlating missense variants with clinical phenotypes and offers a systematic literature review. The use of Whole Exome Sequencing (WES) and in silico tools like SIFT, PolyPhen, and CADD to assess variant pathogenicity is robust and current. A systematic literature review is a strong approach for contextualizing findings.
Limited Functional Validation:
While the paper hypothesizes that missense variants cause a loss-of-function effect, no experimental functional validation (e.g., cell-based or animal models) is performed for the identified Met301Ile variant.
Suggestion: Include a brief discussion of the need for functional validation or outline potential experiments for future studies.
Literature Review Limitations:
Although the review compiles data from existing studies, there is limited critical analysis of discrepancies or gaps in findings between studies.
Suggestion: Provide a deeper critique or synthesis of the reviewed literature, identifying contradictions or uncertainties.
Redundancy in Results and Discussion:
Some sections in the Results (e.g., Table 1 description) are repeated in the Discussion without adding new insights.
Suggestion: Condense redundant data descriptions in the Discussion and instead focus on interpreting the results.
Impact of Dual Diagnoses:
For Patient 2, the dual molecular diagnosis (MMP21 and SCN1A variants) is briefly mentioned, but its clinical impact is not sufficiently discussed.
Suggestion: Explore the potential interplay between the SCN1A variant (epilepsy) and the MMP21 variant to provide a holistic view of the patient's phenotype.
Figures and Visualizations:
While Figure 2 is informative, the 3D protein structure representation could benefit from clearer annotations or zoom-ins on key regions (e.g., Met301).
Suggestion: Enhance the figure to better highlight the role of the catalytic domain in pathogenicity.
Lack of Control Data:
The study does not discuss population-level prevalence of MMP21 variants in unaffected individuals beyond the gnomAD data.
Suggestion: Address whether similar variants are found in asymptomatic individuals and discuss their possible role as risk modifiers rather than definitive pathogenic variants.
Language and Editing:
Minor grammatical inconsistencies and long sentences reduce readability in certain sections.
Suggestion: Simplify complex sentences for clarity and ensure consistency in terminology (e.g., use of "heterotaxy" vs. "situs ambiguus").
Conclusion Section:
The conclusion reiterates the findings but lacks a forward-looking perspective.
Suggestion: Highlight the clinical implications of findings (e.g., inclusion of MMP21 in diagnostic panels) and outline future research directions.
Author Response
Please, see the attached file.

Reviewer 2 Report
Comments and Suggestions for Authors
We thank the authors for their efforts to report 2 interesting cases of patients with the same MMP21 Met301Ile mutations causing cardiac anomalies without extracardiac laterality defects. They add a review of the 17 additional previously reported missense variants.
This article will be helpful to clinical practitioners who want to better understand the results of the WES they prescribe.
We're still a long way from understanding why extracardiac anomalies are or aren't associated with the disease, but as the authors have said, only functional studies will enable us to make progress on this subject.
Author Response
Please, see the attached file.

Round 2
Reviewer 1 Report
Comments and Suggestions for Authors
Accept in present form